# Indoor Environmental Comfort Assessment of Traditional Folk Houses: A Case Study in Southern Anhui, China

**DOI:** 10.3390/ijerph20043024

**Published:** 2023-02-09

**Authors:** Chao Pan, Yunfa Wu, Sarula Chen, Yang Yang

**Affiliations:** 1School of Architecture & Urban Planning, Anhui Jianzhu University, Hefei 230601, China; 2State Key Laboratory of Green Building in Western China, Xi’an University of Architecture & Technology, Xi’an 710055, China; 3Sino-Portugal Joint Laboratory of Cultural Heritage Conservation Science, Gold Mantis School of Architecture, Soochow University, Suzhou 215127, China; 4College of Architecture and Art, Hefei University of Technology, Hefei 230601, China

**Keywords:** southern Anhui, traditional residential houses, indoor environmental quality, human comfort, field survey

## Abstract

Due to the unique geographical location and historical culture, the traditional houses in the southern region of Anhui Province, China (South Anhui) have different indoor environments. In summer and winter, this study adopted a field survey, questionnaire survey, and statistical analysis to carry out a comprehensive field survey on Xixinan Village in South Anhui, and selected a typical traditional residence in the village to evaluate its indoor environment status. The final results show that the overall indoor environment of the traditional houses in South Anhui was awful, including the indoor thermal environment, with high temperature and humidity in summer and cold and humidity in winter. Additionally, the indoor light environment with dim light still had much room for improvement, while the indoor air quality and sound environment were relatively excellent. In addition, this study determined that the neutral temperatures of residents are 15.5 °C and 28.7 °C in winter and summer, respectively, and the comfort range of indoor light intensity is 752.6–1252.5 lx, which determines the adjustment range of indoor environmental parameters for residents’ comfort needs. This paper’s research methods and results provide a reference for the study of residential indoor environments in other regions with similar climatic conditions as South Anhui, and a theoretical basis for architects and engineers to enhance the indoor environment of traditional houses in this region.

## 1. Introduction

In recent years, with the continuous advancement of urban and rural integration in China, the activation and renewal of traditional buildings became particularly important [1,2,3,4]. How to create a low-carbon, comfortable, and livable indoor environment while adhering to the people-oriented and satisfying living use function is the critical point in the revitalization and renewal of traditional buildings [5,6,7,8]. The southern part of Anhui Province, China, surrounded by rivers and mountains, is a bidirectional unfavorable climate zone with hot summers and cold winters. Due to the influence of the economy, technology level, and lifestyle, the range of human comfort of residents in traditional villages in this area is different from other areas [9,10,11]. However, the existing residential research results are not fully applicable to the traditional houses in South Anhui, which have a particular geographical location, unique spatial form, and regional characteristics [12,13]. Therefore, an in-depth assessment of the indoor status and environmental comfort of traditional dwellings in this area is urgent [14].

Experts and scholars conducted the following studies on the effects of indoor environmental comfort on occupants’ health, quality of life, and productivity [15]. Li et al. [16] concluded that a comfortable and slightly warmer ambient temperature range in winter bedrooms may help improve human respiratory immunity. Katrin et al. [17] evaluated the effects of temperature and hot atmospheric conditions on all-cause and cardiovascular mortality in Bangladesh and concluded that high temperatures and heat waves increase the risk of pyrexia and heat-related health injuries in outdoor workers. Michael A et al. [18] concluded that thermal discomfort could contribute to developing chronic diseases, such as respiratory and renal diseases, diabetes, and cardiovascular diseases. Additionally, the level of pollutants in the air can likewise affect users’ health [19]. Xue et al. [20] concluded that the residents of a severely polluted city had high health risks associated with exposure to PM2.5 that exceeded the daily limits of 25 μg/m^3^. He et al. [21] concluded that IAP caused by using coal stoves and Chinese Kang in rural areas was also associated with adverse respiratory health effects in healthy young people through questionnaires, urine collection, and spirometry during the winter holidays. The light environment can also play an essential role in personnel comfort and productivity [22]. Paola et al. [23] studied the quality of the indoor environment in classrooms at the university of Pavia, Italy, through questionnaire research and field monitoring, and concluded a high correlation between average measured illuminance values and perceived visual comfort. Stephen et al. [24] concluded that correctly using indoor and outdoor lighting and the color is vital for human and ecosystem health. In addition, long-term exposure to high noise levels can be hazardous to human health and may cause hearing damage, hypertension, and even cardiovascular injury [25]. Hypertension is a significant independent risk factor for events such as myocardial infarction and stroke. David et al. [26] concluded that nocturnal noise was positively associated with diastolic blood pressure and an increased risk of hypertension was associated with ambient noise. Fabrizio et al. [27] investigated the relationship between noise judgment in schoolgoers aged 11–18 and noise measurements aimed at evaluating their exposure at school. The results show that noise perception and consequent disturbance are highly correlated with classroom acoustics, and confirm that annoyance represents the most widespread subjective response to noise.

In conclusion, the research related to the indoor environment mainly focuses on modern houses, offices, schools, moving stations, and shopping centers [28,29,30,31,32,33]. Secondly, the research on the indoor environmental comfort of traditional houses mainly focuses on the single thermal environment, such as indoor temperature and humidity. It lacks research on overall indoor environmental comforts, such as indoor light environment, air quality, and sound environment [34,35,36,37,38]. Due to the regional differences and the unique characteristics of the buildings, the existing research needs to provide better guidance for revitalizing and upgrading the traditional houses in South Anhui. Therefore, assessing the contemporary interior of traditional houses in this area is urgent.

Therefore, the current situation of the indoor thermal environment, light environment, and indoor air quality of traditional residential houses in the ancient village of Xixinan, Huizhou District, Huangshan City, Anhui Province, China, was comprehensively investigated, and a typical traditional residence in Yao’s House was taken as an example. Through the methods of subjective perception evaluation (questionnaire survey), objective data analysis (field monitoring), and statistical analysis, this paper conducts an in-depth and comprehensive survey of the indoor environment of traditional residential buildings and synthesizes its current situation and the feedback results of residents’ subjective comfort, to determine the adjustment range of indoor environment parameters that can meet residents’ comfort needs. In chapter 2, the survey of villages and the measurement of traditional dwellings are introduced in detail. The research protocol, instruments used, questionnaires, and comfort assessment methods are described in chapter 3. Chapter 4 analyzes the results of the survey and questionnaire. Chapter 5 summarizes the discussion results, hoping to enrich and improve the research theory of traditional residential indoor environment comfort and promote the renewal of regional residential buildings.

## 2. Materials and Methods

### 2.1. Research Steps

This paper consists of four steps (Figure 1). First, the basic background information of existing traditional villages in Huangshan City is summarized and investigated by reading literature and other methods (Step 1). Secondly, according to the main gathering villages of traditional dwellings, the ancient village of Xixinan, Huizhou District, Huangshan City, Anhui Province, China, and a typical traditional residence located in the village, “Yao’s House”, are selected as the objects for monitoring the indoor environment data of traditional dwellings. Step 3: Conduct the actual survey, field measurement, and questionnaire survey on the existing traditional dwellings in Xixinan Village, and finally, analyze and make statistics on them by using several evaluation indexes, such as APMV, indoor light intensity, and daylight factors. See details in Section 2.3, Section 2.4, Section 2.5 and Section 2.6.

### 2.2. Research Subjects

Anhui Province (Figure 2) is located in China’s hot summer and cold winter zone, and due to its topography and other conditions, it is divided into three regions: southern, central, and northern Anhui, each of which has its climate characteristics [39,40]. Among them, the characteristics of hot and high humidity, cold and damp, are the most obvious in southern Anhui, and its comprehensive demand for heat, cold, and humidity prevention is stronger [41]. In addition to Xin’an River, there is also Qingge River, which originates from the northern slope of Huangshan Mountain, and there are Huangshan Mountain, Qiyun Mountain (Baiyue), and its remaining veins, which is a mountainous area called “eight mountains and one water and one field” [42]. The Huizhou traditional houses located in the southern part of Anhui reflect the characteristics of the mountainous area, feng shui will, and the tendency of regional beauty decoration, and their structure is the same. There are multiple courtyards inside the residences. Folk houses generally sit on the north to the south, according to the terrain built [43]. The layout of traditional houses in Huizhou is the same, with symmetrical distribution in the central axis. The courtyard in front of the hall is called “Patios”, which has the function of lighting and ventilation and has the auspicious meaning of “four waters return to the hall” [44].

This paper selects the ancient village of Xixinan, which is located in the Huizhou District of Huangshan City, as the research object. Xixinan Village was built in the Tang Dynasty, with a history of more than 1200 years, and the village now has more than 10 well-preserved Ming Dynasty buildings and more than 100 Qing Dynasty dwellings, which is a typical gathering village of traditional dwellings in southern Anhui [45]. The traditional houses in Xixinan Village are usually compact in group layout, with wooden beams and columns as the main load-bearing structure and brick, stone, and earth as the main wall materials. According to the location and layout of the patio, there are three types (Figure 3): “凹” style, “回” style, and “日”. In this study, the “Yao’s House” in Xixinan Village was selected as the object of real-time monitoring of indoor environmental data, and its unique “日” shape layout is a typical layout of traditional residential buildings in southern Anhui [46,47,48]. The dwellings face north and south, with an axisymmetric layout, 9 m in width and 23.3 m in depth. There are two patios of different scales in the dwelling, which are relatively intact and representative of the study.

### 2.3. Evaluation Method of Indoor Environmental Comfort

The content of this study is the comprehensive indoor environmental comfort of traditional houses, and the comfort evaluation hierarchy includes the comprehensive comfort evaluation target, human comfort standard layer, and data index layer (Table 1).

#### 2.3.1. Thermal Environment Comfort Assessment Method

ASHRAE defines thermal comfort as a thermal environment in which people feel psychologically satisfied [50]. Thermal comfort consists of four indoor climate factors (air temperature, air humidity, average radiation temperature, and airflow velocity) and two human factors (human metabolic rate and clothing thermal resistance).

(1)*PMV*-PPD and *aPMV* thermal comfort evaluation

The predicted mean vote (*PMV*) proposed by Prof. Fanger is one of the most widely used indicators, which can reflect the relationship between the factors more comprehensively and objectively [51]. Accordingly, *PMV* is expressed as follows.
(1)$$\begin{array}{cc} PMV=[0.303\; & exp\left(-0.036M\right)+0.0275]\\ & \times \{M-W-3.05\left[5.733-0.07\left(M-W\right)-P_{a}\right]-0.42\left(M-W-58.2\right)\\ & -0.0173M\left(5.867-P_{a}\right)-0.0014M\left(34-t_{a}\right)-3.96\times 10^{-8}f_{cl}\left[{\left(t_{cl}+273\right)}^{4}-{\left(\bar{t_{r}}+273\right)}^{4}\right]\\ & -f_{cl}h_{cl}\left(t_{cl}-t_{a}\right)\} \end{array}$$
where *M* is the metabolic rate, *W* is the effective mechanical power, and this value is taken as 0 in this study, *t_a_* is the room air temperature, and *t_r_* is the average radiation temperature. Where *f_c_*_l_ is the clothing thermal resistance and *t_cl_* is the outer surface temperature of the garment, which can be calculated by the following functions:(2)$$f_{cl}\{\begin{array}{c} 1.00+1.290I_{cl}\;\;\;for\;\;\;I_{cl}\leq 0.078m^{2}\cdot \frac{K}{W}\\ 1.00+0.645I_{cl}\;\;\;for\;\;\;I_{cl}\geq 0.078m^{2}\cdot \frac{K}{W} \end{array}$$
where *v* is the airflow rate, *P_a_* is the partial pressure of water vapor, and the calculation function is as follows:(3)$$P_{a}=1000\cdot \left(RH\right)\cdot exp\left(18.67-4030.18\right)/\left(t_{a}+235\right)$$
where *RH* stands for relative humidity.

However, the predicted mean vote (*PMV*) is more applicable to the artificial thermal environment indoors. Since most of the traditional residential houses in southern Anhui use natural ventilation to adapt to the summer and winter climate, the evaluation index of non-artificial heat and cold sources (*aPMV*) in the evaluation standard for indoor thermal environment in Civil Buildings of China is chosen to evaluate the indoor thermal environment to reflect the average indoor thermal sensory index more accurately [49].
(4)aPMV=PMV/(1+λ)
where *λ* is the coefficient of adaptation. When *PMV* ≥ 0 is, *λ* = 0.21; and when *PMV* ≤ 0 is, *λ* = −0.49.

(2)Operating temperature

The operating temperature *t_op_* reflects the effect of *t_a_* and *t_r_* on the body’s thermal sensation, *t_op_* is also often used as an indicator for evaluating indoor thermal environments and is expressed as follows:(5)$$t_{op}=\left(t_{a}+t_{r}\right)/2.\;$$

#### 2.3.2. Light Environment Comfort Assessment Method

The research found that the side windows do not show dazzling light due to the shading of the eaves of traditional houses and the long side windows. Therefore, this paper’s main indoor light environment evaluation index is mainly based on the natural light illumination value and the lighting coefficient. Since the lighting coefficient is converted from the illumination value, the illumination value is used as the main measurement index in this study.

#### 2.3.3. Indoor Air Quality Evaluation Methods

The CO_2_ concentration generated by human activities is often used to characterize the freshness of indoor air’s freshness or ventilation strength. The content of PM_2.5_ per cubic meter of air represents the degree of air pollution [52]. Therefore, PM_2.5_ and CO_2_ concentration distribution are used to evaluate indoor environmental quality comfort.

#### 2.3.4. Sound Environment Comfort Assessment Method

Noise is a subjective evaluation standard. That is, all the sound that affects others is noise [53]. Because the results measured by A-weighted sound pressure are similar to the human ear’s perception of the loudness of sound, A-weighted sound pressure level is one of the most widely used indicators in noise measurement in the world. It became one of the main indicators for evaluating indoor environmental noise in the International Organization for Standardization and most countries [54,55,56]. Previous studies found that the equivalent continuous A-weighted sound pressure level (*L_Aeq_*) is strongly correlated with sound comfort, so the noise evaluation index in this study is *L_Aeq_* [57,58].
(6)$$L_{Aeq}=10\mathrm{lg}\left(\frac{1}{T}\int_{0}^{T}10^{0.1\times L_{A}}dt\right)$$
where *L_A_* is the instantaneous A-weighted sound pressure level at time t, *T* is the specified measurement period.

### 2.4. Questionnaire Research

In this paper, the questionnaire study and field monitoring were conducted on typical meteorological days in winter (25 to 28 January 2022) and summer (8 to 13 July 2022) in Huangshan City. The details are as follows.

The contents of this questionnaire study are as follows:(1)Collect basic information about the respondents and their traditional houses.(2)Respondents’ comfort ratings of indoor thermal environment, light environment, indoor air quality, and sound environment.(3)Adaptive behavior of the respondents in adapting to changes in the outdoor environment.

The research questionnaire is shown in the appendix. The seven-point scale specified in Appendix E of ASHRAE-55 (Figure 4) was used to evaluate subjective human comfort under different environmental factors. The range of environmental comfort evaluation was from −3 to 3 [50]. To achieve a stable state, all respondents sat in their rooms in a relaxed state for at least 15 min while conducting the questionnaire.

Table 2 presents the basic demographic information of the 359 residents who participated in the questionnaire study during the winter and summer. A total of 677 questionnaires were returned for the questionnaire study, of which, 328 were returned in winter and 349 in summer.

### 2.5. Field Measurements

The field monitoring used the instruments listed in Table 3, and the automatic recording interval of each instrument was set at 5 s. The accuracy of all monitoring equipment was compounded with the requirements specified in ASHRAE 55-2017, GB/T 5699-2017, and GB 50118-2010. The current status of “Yao’s House” (Figure 5) and the arrangement of monitoring points (Figure 6) are as follows:

### 2.6. Statistical Methods

Indoor environmental parameters and human comfort evaluation are generally normally distributed, and if a conventional linear regression model is used, the results will produce some bias. Therefore, the distribution frequency of sample numbers of human comfort feeling voting in each environmental parameter interval is used as the weight of weighted regression model analysis [59].

In this paper, based on the sample size of the distribution interval of indoor environmental parameters in different seasons, the linear regression model of comfort evaluation and indoor environmental parameters was obtained by linear regression analysis [60].

The model for the weighted linear regression is as follows:(7)y=a+bx
where *a* and *b* are linear regression coefficients with the following expressions:(8)$$a=\frac{\left(\sum_{i=1}^{N}w_{i}\right)\left(\sum_{i=1}^{N}w_{i}x_{i}y_{i}\right)-\left(\sum_{i=1}^{N}w_{i}x_{i}\right)\left(\sum_{i=1}^{N}w_{i}y_{i}\right)}{\left(\sum_{i=1}^{N}w_{i}\right)\left(\sum_{i=1}^{N}w_{i}x_{i}^{2}\right)-{\left(\sum_{i=1}^{N}w_{i}x_{i}\right)}^{2}}$$
(9)$$b=\frac{\left(\sum_{i=1}^{N}w_{i}y_{i}\right)-a\left(\sum_{i=1}^{N}w_{i}x_{i}\right)}{\sum_{i=1}^{N}w_{i}}$$
where *x_i_* is the indoor environmental parameter, *y_i_* is the comfort vote corresponding to each environmental parameter, *w_i_* is the sample size of the vote corresponding to each environmental parameter, and *N* is the number of environmental parameter intervals.
(10)$$R^{2}=\frac{\sum_{i=1}^{N}w_{i}{\left(b+ax_{i}-\frac{\sum_{i=1}^{N}w_{i}y_{i}}{\sum_{i=1}^{N}w_{i}}\right)}^{2}}{\sum_{i=1}^{N}w_{i}{\left(y_{i}-\frac{\sum_{i=1}^{N}w_{i}y_{i}}{\sum_{i=1}^{N}w_{i}}\right)}^{2}}$$
where *R*^2^ is the linear correlation coefficient of the regression equation.

## 3. Results

### 3.1. Thermal Environment Comfort

According to the results of the questionnaire research (Figure 7), the degree of satisfaction of the respondents from Xixinan Village about the indoor thermal environment in winter and summer are, respectively, rated as 8.2% and 8.8%, while the degree of dissatisfaction is rated as 75.4% and 70.2%, in which the degree about winter is 5.2% higher than summer. The reasons are that, on the one hand, the open patios in the traditional dwellings with weak air impermeability lead to serving heat dissipation inside. On the other hand, although the traditional house adopts a compact arrangement to reduce heat loss, the exterior envelope is poorly insulated, and the indoor temperature is easily affected by the outdoors, resulting in low satisfaction with the thermal environment of the respondents.

Due to the harsh natural environment, economic conditions, as well as daily living habits of the residents, the heat adaptation behavior of the respondents in Xixinan Village was different from that of residents in other areas. The data from Figure 8 show that respondents in summer and winter were different in thermal adaptation behaviors and usage proportion. When the outdoor climate changes, respondents will replace their clothes as their primary choice to adjust their thermal comfort status. Residents also adopt adaptive behaviors, such as opening and closing doors and windows, drinking hot water, and sunbathing. Because of high operating costs, more than half of the respondents said they only use air conditioners and other equipment to regulate indoor temperature in summer.

According to the dressing conditions of the respondents collected in the field research of this study: the mean value of the clothing thermal resistance of the respondents (2.07 clo) was much higher than the recommended winter dressing standard (1 clo) in ASHRAE 55-2017 on account of the cold and humid indoor winter; in summer, the mean value of the clothing thermal resistance of the respondents was 0.39 clo. The relationship between thermal insulation and indoor operative temperature is shown using raw data in Figure 9. A scatter diagram of operative temperature (*t_op_*_)_ and clothing insulation (*I_cl_*) was plotted to calculate the following linear regression equations:(11)$$\mathrm{Winter}:I_{clo}=-0.040\times t_{op}+2.37\;,\;R^{2}=0.689$$
(12)$$\mathrm{Summer}:I_{clo}=-0.037\times t_{op}+1.52\;,\;R^{2}=0.791.$$

This result demonstrates that clothing insulation is negatively correlated with operative temperature. The slopes of various regression equations are different, so people living in different regions and buildings have diverse needs for clothing warmth. The slope of the linear relationship between the clothing insulation *I_clo_* and operative temperature for respondents in Nanyang City, Henan Province, and Shanghai city, which are also in the hot summer and cold winter zone, are greater than the slope found in this study, which indicates that residents of traditional dwellings have lower thermal sensitivity than those of modern houses.

On the grounds of the indoor thermal environment monitoring results of Yao’s House (Figure 10), the indoor environment was cold and damp in winter, and had high temperature and humidity in summer. The indoor wind environment was in the state of breeze or static wind for a long time. In short, the indoor thermal environment was particularly bad.

The air temperature variation in the house was closely related to solar radiation, and the variation trend of air relative humidity at each measuring point was opposite to that of air temperature (Figure 10). The bedroom faces north so as not to be directly exposed to the sun, and the temperature fluctuates less throughout the day. Its average temperature in winter is 1.8 °C higher than the outdoor measuring point, and in summer, it is 2.6 °C lower. Because the air exchange in the bedroom relies primarily on a small window opening onto the interior patio, resulting in its small effective ventilation area, the room is often breezy or still. Hence, it averages over 75% humidity in winter and summer, exceeding the comfortable temperature and humidity range. The temperature and humidity of the other four measuring points were affected by solar radiation, and the changes were drastic. Compared with other indoor spaces in the dwellings, the patios were continuously exposed to direct sunlight during summer, and the average temperature continued to rise. At the same time, the highest temperature on the patios was 39.6 °C, second only to the outdoors. Although the sun’s radiation accelerated the evaporation of water vapor in the patios, the air humidity was still above 75%. In winter, due to the occlusion formed by the overhanging eaves, the patios were less exposed to direct sunlight. With the increase in the midday solar radiation, the vertical wind pulling effect of the patios strengthened the thermal pressure ventilation effect, and the patios became the main ventilation path, which resulted in a relatively severe heat loss of the house. In addition, the average temperature of the patios in winter was the lowest among the monitoring points, only 7.7 °C.

Yao’s House’s poor indoor thermal environment was also reflected in the monitoring results of indoor comfort (Figure 11). During the monitoring period in winter and summer, the *aPMV* values in all periods hardly reached the thermal comfort range (−1, 1) specified in the evaluation standard for indoor thermal and humid environments in civil buildings [49]. In particular, *aPMV* values were lower in winter, with values below −1.0 even at higher daytime temperatures, while in summer, all daytime *aPMV* values were above 1.0, which indicates that the thermal environment inside the house was very uncomfortable. The raw data used in Figure 11 show the relationship between the thermal predicted mean vote and operating temperature. A scatter plot of the thermal predicted mean vote (*aPMV*) versus operating temperature (*t_op_*) was plotted to calculate the linear regression equation.
(13)$$\mathrm{Winter}:\;aPMV=0.21\times t_{op}-3.26\;,\;R^{2}=0.801$$
(14)$$\mathrm{Summer}:\;aPMV=0.16\times t_{op}-4.60\;,\;R^{2}=0.891.$$

The *aPMV* value was set to 0, and the neutral operating temperature of traditional house residents in Xixinan Village in winter and summer was 15.5 °C and 28.8 °C. In Shanghai, which is also in the cold winter and hot summer zone, the neutral operating temperature of modern house residents in winter was 20.8 °C, which was 5.3 °C different from that of residents in Xixinan Village. While in summer, the neutral operating temperature was 20.8 °C, which was lower than that of traditional house residents (28.8 °C) [61].

### 3.2. Indoor Light Environment Comfort

Based on the consequences of the indoor light environment questionnaire, the proportion of respondents who rated it as the comfort level in winter (14.4%) was much lower than that in summer (57.0%), which indicates that the indoor light environment in summer was better than that in winter (Figure 12).

Owing to the influence of historical and humanistic factors, the traditional dwellings in south Anhui had tall exterior walls and few windows on the facade. Even if they were open, they were usually small windows. According to the residential design code, the minimum requirement for the ratio of windows to the floor in bedrooms and living rooms is 1/7. According to the survey results, this study finds that the ratio of windows to floor in traditional houses in Xixinan Village was only 49.01% higher than this minimum [62].

During the research, this study found that Yao’s House’s interior decoration was timeworn, and the interior was unmaintained for a long time, resulting in the meager reflection ratio of interior surface finishing materials (Figure 13). Compared with the reference value of the suitable reflectance ratio of finish materials in the design standards for building light of China, it was at a low threshold (see Table 4) [63]. Depending on the monitoring outcomes of the indoor light environment of Yao’s House (Figure 14), the values that met the lower limit of indoor natural light illumination (300 lx) stipulated in the standards for architectural lighting design of China in winter and summer were only 19.5% and 28.3%. The values that met the lower limit of the natural light coefficient (2%) stipulated in the regulations were only 28.6% and 15.6%, which indicated that the natural lighting of the dwellings could be better. Among them, the bedroom had the worst indoor light environment. While the maximum light intensity of the indoor terrace was 7059.3 lx, that of the bedroom was only 19.9 lx.

The average comfort vote of indoor natural light intensity was calculated, and the relationships between natural indoor illumination (*L*) and light environment comfort vote (*I_L_*) were given in Figure 15. The fitting formulas are provided as follows:(15)$$I_{L}=0.004\times L-2.01\;,\;R^{2}=0.727$$
which indicates a significant correlation between the comfort level of the light environment and natural light intensity. The quantity interval (1,3) in evaluating the subjective comfort of the light environment can be expressed as light comfort. Through the linear fitting equation calculation, the comfort interval of the indoor natural light intensity of traditional houses in Xixinan Village was finally obtained as 752.6–1252.5 lx.

### 3.3. Indoor Environmental Quality

According to the results of the indoor environmental quality questionnaire (Figure 16), the majority of respondents in winter (21.5%) rated the indoor environmental quality of traditional houses as slightly more uncomfortable than in summer (20.6%).

According to the indoor environmental quality monitoring results of Yao’s House (Figure 17), indoor CO_2_ concentration in the dwellings in winter ranged from 449.4 to 541.6 ppm, and the values mainly ranged from 460.4 to 480.6 ppm. The PM_2.5_ concentration ranges from 4 to 58 μg/m^3^, and the values are mainly between 5 and 27 μg/m^3^, while in the summer, the indoor CO_2_ concentration in Yao’s House ranged from 403.8 to 520.7 ppm, and the values mainly ranged from 422.2 to 477.5 ppm. The PM_2.5_ concentration ranges from 7 to 66 μg/m^3^, and the values are mainly between 13 and 27 μg/m^3^. In the monitoring process, residents cooked at 6:00–7:00, 11:30–12:30, and 17:30–18:30, so the value fluctuated in these periods. During the monitoring period, the indoor CO_2_ and PM_2.5_ concentrations in Yao’s House were both lower than the upper limits specified in the indoor air quality standards of China (CO_2_ concentration of 1000 ppm and PM_2.5_ concentration of 75 μg/m^3^) [64].

### 3.4. Indoor Sound Environment

According to the results of the indoor sound environment questionnaire, 70.8% of respondents felt comfortable with the indoor sound environment, which indicated that the respondents had a high degree of satisfaction with the sound environment (Figure 18).

According to Yao’s House sound pressure level monitoring results during daytime and nighttime (Figure 19). At the period of 8:30–9:20, the maximum outdoor sound pressure level was 72 dB(A), while the indoor values were much smaller than the outdoor ones, with an indoor *L_Aeq_* value of 45.9 dB(A). In addition, the maximum indoor sound pressure level appeared in the hall, with a value of 58.6 dB(A), mainly due to the indoor people’s frequent activities. When it came to 10:00, the indoor and outdoor sound pressure levels weakened, and the indoor *L_Aeq_* value was 40.1 dB(A). Furthermore, the peak indoor sound pressure level still appeared in the hall, with a value of 61.3 dB(A). According to the sound environment quality standard GB3096-2008, the traditional dwellings in Huizhou Province implement the Class 1 standard, with the equivalent continuous sound pressure level limit of 55 dB(A) during daytime and 45 dB(A) at nighttime [65]. The Yao’s House met the equivalent continuous sound pressure level limits specified in the standard during the monitoring time.

## 4. Discussions

### 4.1. Thermal Environment

According to the results of the indoor thermal environment survey, the current thermal environment of traditional houses in southern Anhui is deplorable. Since the air exchange in traditional houses mainly depends on the patio, which cannot form good ventilation and shading effects in summer and dissipates a lot of indoor heat in winter. It leads to a cold and humid winter interior and high summer temperatures and humidity. Therefore, residents can determine a reasonable window-to-wall ratio according to the function of each room under the premise of ensuring normal ventilation and lighting in traditional residential houses. The research results show that when the window-to-wall ratio is 0.4, the indoor energy consumption of the house is lower while getting good lighting and ventilation [66]. Secondly, because the exterior walls of traditional houses in southern Anhui are in disrepair, the heat insulation performance of the broken outer envelope is poor, affecting the indoor temperature and humidity conditions more affected by the outdoors. Therefore, applying foam concrete and inorganic interior wall insulation mortar is an excellent choice to enable walls to meet insulation requirements while maintaining a traditional appearance [67]. It is a good choice in areas with low industrialization of traditional houses. Based on the neutral temperature results of residents in winter and summer, the respondents were primarily middle-aged and older people, over 50 years old, who had lower thermal sensitivity, resulting in a lower psychological expectation of indoor ambient temperature. Second, the adaptive behaviors adopted by residents due to their prolonged exposure to the poor thermal environment greatly alleviates the discomfort that residents experience during climate change. They have better tolerance for the harsh environment than urban residents. Therefore, when designing heating systems for residents, the engineers can set the indoor temperature in their thermal comfort range.

### 4.2. Light Environment

As far as the indoor light environment is concerned, traditional houses in southern Anhui could be better, and residents are less satisfied with the indoor light environment. The results show that because the rooms of traditional dwellings of South Anhui have small or no windows, the indoor lighting mainly depends on the patios, which lead to a small indoor lighting area and uneven distribution of light intensity. Therefore, in addition to auxiliary spaces, such as attics and stairwells, which can be designed with top lighting, some rooms with larger areas can also use top lighting, of which opening skylights and laying bright tiles are the most common techniques. In addition, residents can confirm a good window opening area by calculating the window-to-land ratio and adjusting accordingly to the lighting demand of functional rooms. The research found that the complex window pane system of traditional houses in southern Anhui blocked most of the natural light from entering [14]. Therefore, when optimizing the indoor light environment, residents can select the appropriate window pane structure by combining the overall style characteristics of the building and the practical use needs of the rooms, prioritizing the simple design with a relatively sizeable virtual reality [68]. According to the statistical analysis results, the comfort level of the indoor light environment of traditional houses in south Anhui has a linear relationship with light intensity, and the comfort range of indoor light intensity is 752.6–1252.5 lx. Therefore, residents can appropriately use a combination of general, local, and mixed lighting to improve indoor light intensity and illumination uniformity while reducing lighting energy consumption.

### 4.3. Indoor Environmental Quality

Regarding air quality, the indoor CO_2_ and PM_2.5_ concentrations are within the standard requirements. Since the main air exchange of traditional houses in southern Anhui depends on the patios and the windows on the facade, the windows on the facade are usually small or even windowless. The tall external walls of the houses make the thermal pressure ventilation of the patio less efficient. The smoke exhaust facilities of the traditional dwellings work less efficiently, so the indoor pollutants cannot be discharged quickly to the outdoors. The indoor CO_2_ and PM_2.5_ concentration monitoring data rise briefly. The purpose of indoor air quality improvement and renovation for traditional houses in southern Anhui is to enhance natural ventilation. Therefore, in the design of ventilation renovation, in addition to the space design to effectively improve the efficiency of indoor natural ventilation, residents can also use mechanical ventilation to achieve a better indoor environment [69].

### 4.4. Sound Environment

In terms of sound environment, the sound insulation effect is good because of the high exterior walls and windowless facade of traditional houses in southern Anhui. In addition, the overall sound environment of the village is excellent because the noise sound level is low in Xixinan Village, which is far away from the urban area. The short-term high-pressure noise sources mainly belong to the resident’s daily life and labor and do not have much impact on the residents’ daily life during the day and rest at night. Therefore, when optimizing the indoor sound environment of traditional houses in southern Anhui, we can arrange suitable insulation materials on ceilings and walls and choose doors and windows with better sound insulation performance to improve indoor sound insulation performance [70]. In addition, the floor plan should be divided into dynamic and static zones, with rooms requiring quietness set on the back side of the noise and noise-generating locations, such as kitchens and bathrooms arranged centrally and as far away from bedrooms as possible [71].

### 4.5. Outlook

Due to the limited time to obtain the approval of the householder for monitoring, the limited content of space, and the number of environmental monitoring equipment, the subsequent study will use numerical simulation to compare and analyze the indoor climate of traditional houses in southern Anhui with different plan shapes, thus filling the gap in the study of traditional houses. These questions can be further explored in future research.

## 5. Conclusions

This study systematically analyzes the current situation of the indoor environment of traditional dwellings in South Anhui through a combination of the questionnaire survey, on-site monitoring, and statistical analysis, taking “Yao’s House” as an example. This study provides a new method for the design and post-use evaluation of the indoor environment parameters of the traditional houses in South Anhui, and can provide strategies for optimizing the indoor environment of different types of traditional houses in this area based on the data results. The main conclusions can be summarized as follows:In terms of the indoor thermal environment, the satisfaction of residents with the indoor thermal environment in winter and summer is particularly low, only 8.2% and 8.8%. In addition, the percentage of *aPMV* values meeting the standards in winter and summer was only 5.2% and 8.0%. Meanwhile, the neutral temperature of residents in winter and summer was 15.5 °C and 28.7 °C, respectively, which implies that they have high adaptability to the harsh environment.In terms of indoor light environment, the proportion of respondents satisfied in winter and summer was 14.4% and 57.0%, respectively. In addition, only 19.5% and 28.3% of indoor natural illumination in winter and summer are higher than the lower limit (300 lx) specified in the standard, and only 28.6% and 15.6% of the natural light-harvesting coefficient are higher than the lower limit (2%). Finally, according to the linear relationship between the comfort of the indoor light environment and the light intensity of traditional houses in South Anhui, the locals’ comfort range of indoor light intensity is 752.6–1252.5 lx.In terms of indoor air quality, traditional dwellings in South Anhui performed well. In winter and summer, 46.1% and 66.6% of the respondents were comfortable with indoor air quality, respectively. More importantly, the concentrations of CO_2_ and PM_2.5_ indoors in Yao’s House in winter and summer met the regulations in the standard during the monitoring period.Respondents of traditional residential houses in South Anhui were highly satisfied with the indoor sound environment, with a satisfaction rate of 70.8%; during the monitoring period, the indoor daytime and nighttime sound pressure level values of Yao’s House all meet the standard. The transient high sound pressure noise sources appearing indoors and outdoors are all from the residents’ living and production work and do not cause much impact on the resident’s daily life and rest.

This study provides a theoretical basis for the improvement of the indoor environment of traditional houses in South Anhui, and also provides a reference for the study of the indoor environment of other houses with the climatic conditions of this region. In addition, the research methods, evaluation indexes, and analysis methods of this study apply to the study of the indoor environment of traditional houses in other areas. However, due to the unique geographical location and climatic environment of South Anhui, the research results of this paper do not apply to residential studies in other climatic regions, such as severe cold regions.

## Figures and Tables

**Figure 1 ijerph-20-03024-f001:**
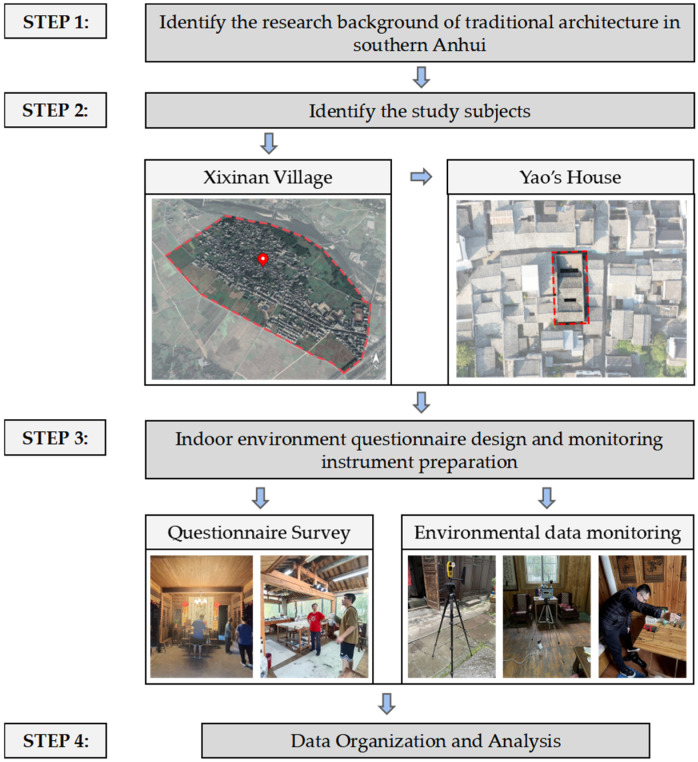
Research steps.

**Figure 2 ijerph-20-03024-f002:**
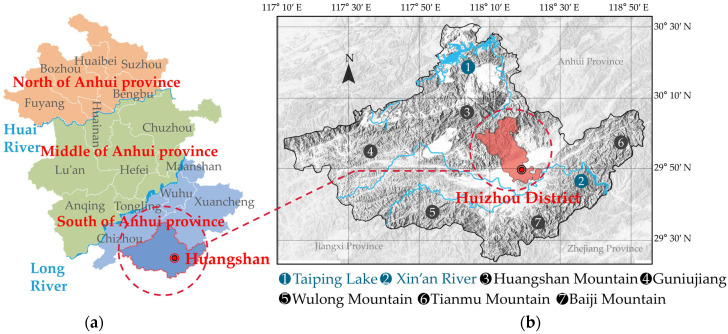
Huizhou District location map. (**a**) Location of southern Anhui and Huangshan city. (**b**) Map of Huangshan City and location of Huizhou District.

**Figure 3 ijerph-20-03024-f003:**
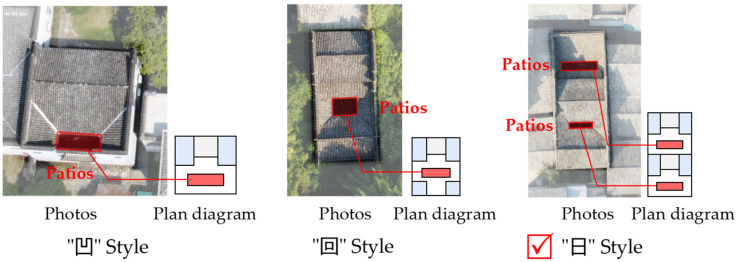
Three typical floor plans of traditional houses in Xixinan Village (The "凹", "回" and "日" represent the building types).

**Figure 4 ijerph-20-03024-f004:**
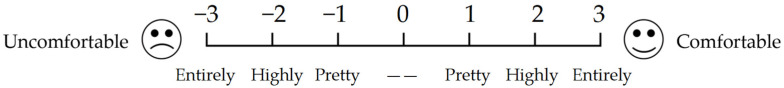
Semantic difference scale for comfort rating scale.

**Figure 5 ijerph-20-03024-f005:**
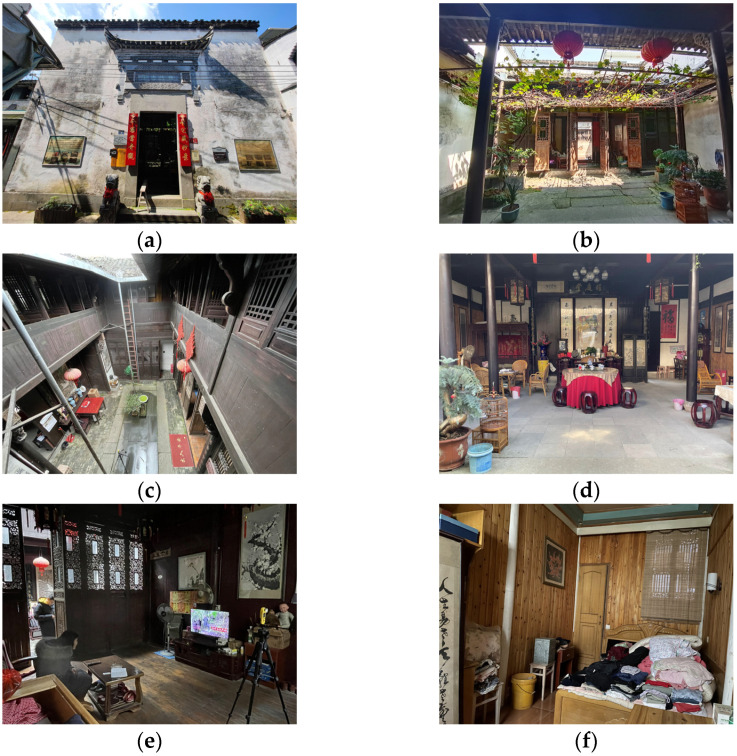
Photographs of the interior of the “Yao’s House”. (**a**) Building front elevation. (**b**) Front patio. (**c**) Hall. (**d**) Rear patio. (**e**) Rear hall. (**f**) Bedroom.

**Figure 6 ijerph-20-03024-f006:**
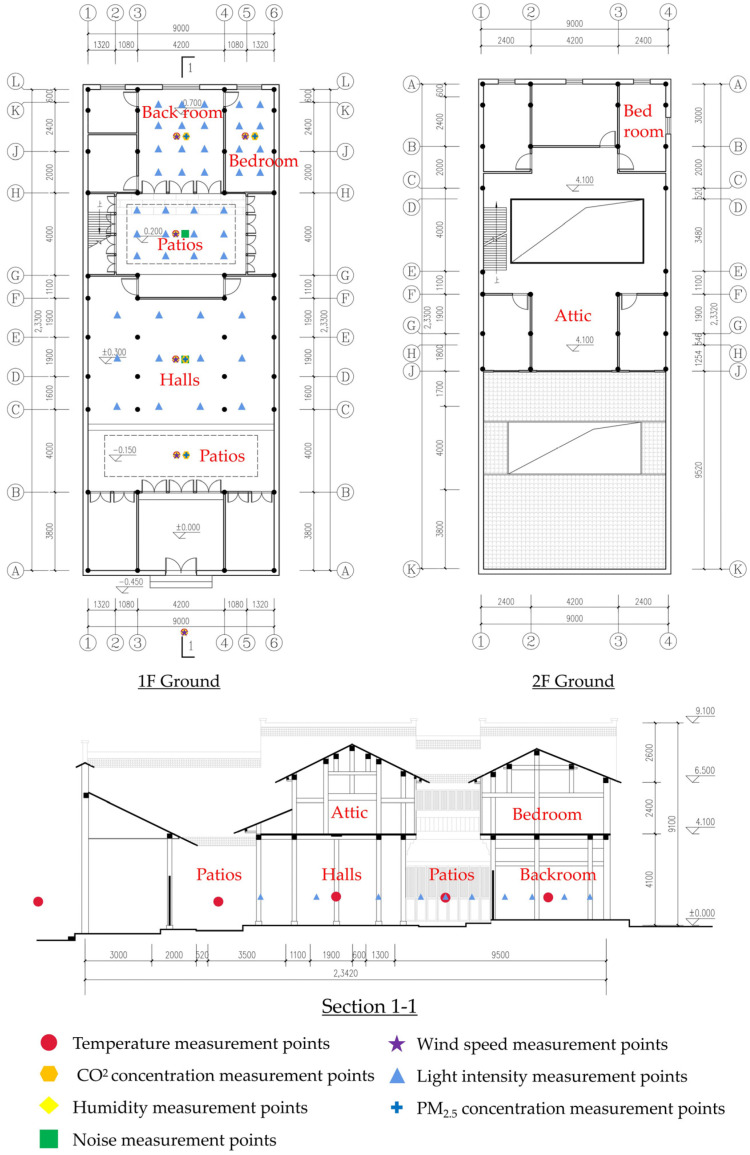
The layout of the “Yao’s House” and the location of the monitoring points.

**Figure 7 ijerph-20-03024-f007:**
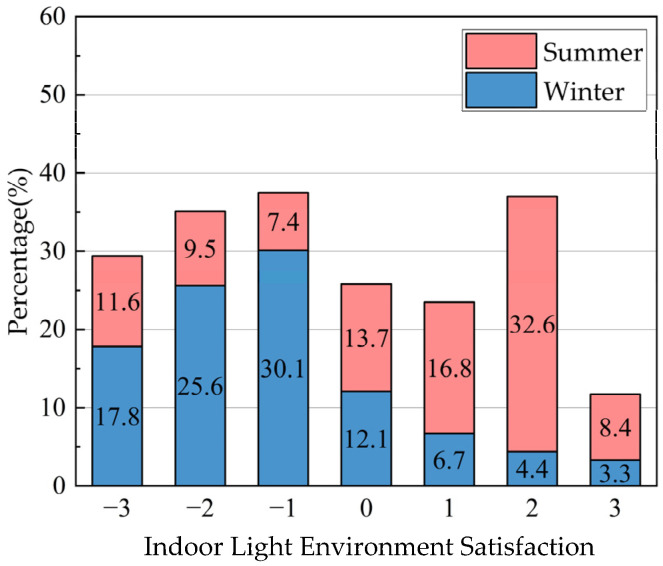
Human subjective comfort evaluation—indoor thermal environment.

**Figure 8 ijerph-20-03024-f008:**
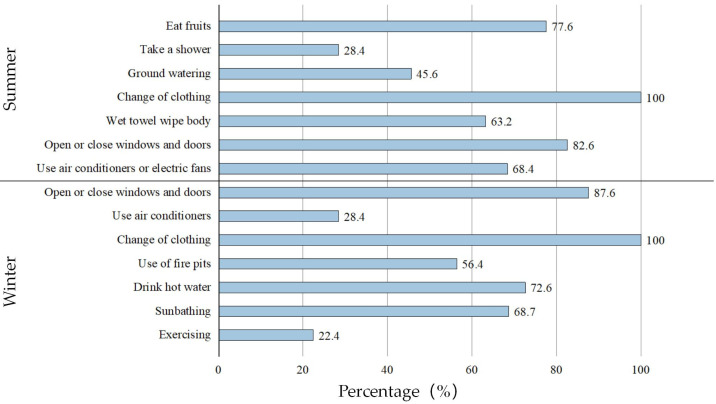
Statistics of residents’ thermal adaptive behavior.

**Figure 9 ijerph-20-03024-f009:**
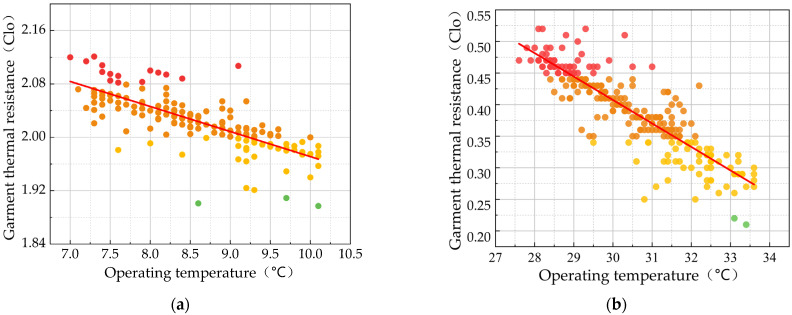
Relationship between the indoor operating temperature and thermal resistance of clothing in traditional residential houses in Xixinan Village. (**a**) Winter. (**b**) Summer.

**Figure 10 ijerph-20-03024-f010:**
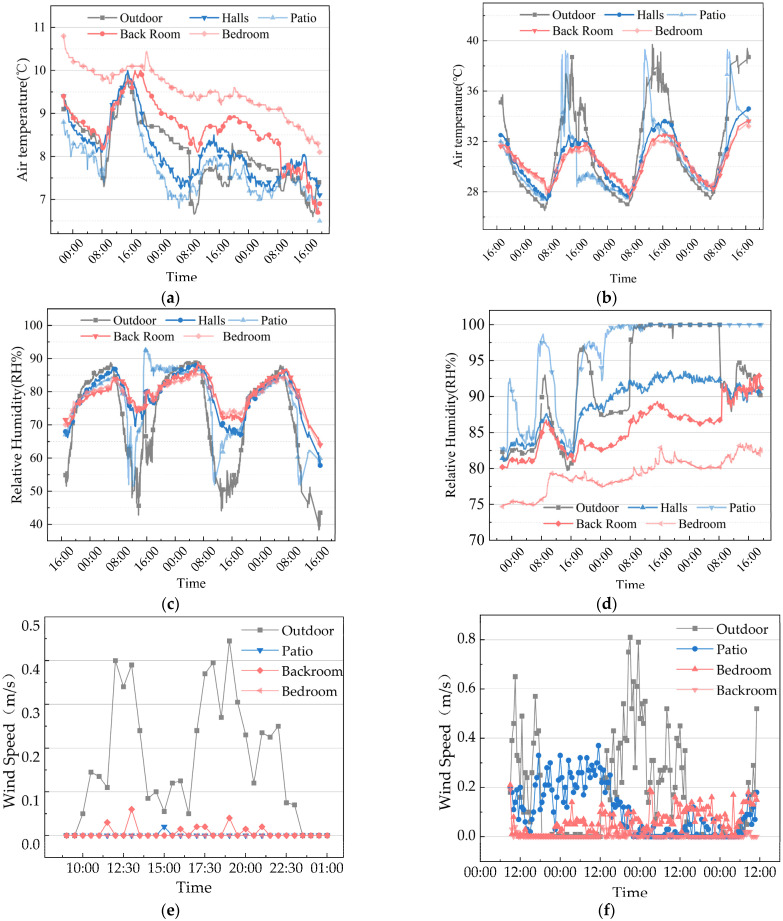
Indoor thermal environment monitoring results of the “Yao’s House”. (**a**) Indoor air temperature distribution in winter; (**b**) indoor air temperature distribution in summer; (**c**) indoor relative humidity distribution in winter; (**d**) indoor relative humidity distribution in summer; (**e**) indoor air velocity distribution in winter; and (**f**) indoor air velocity distribution in summer.

**Figure 11 ijerph-20-03024-f011:**
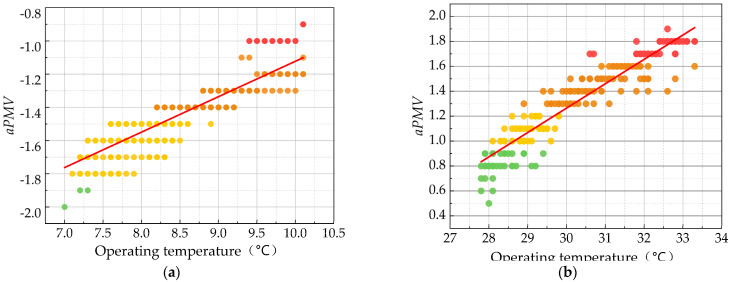
Operating temperature as a function of aPMV distribution. (**a**) Winter; (**b**) summer.

**Figure 12 ijerph-20-03024-f012:**
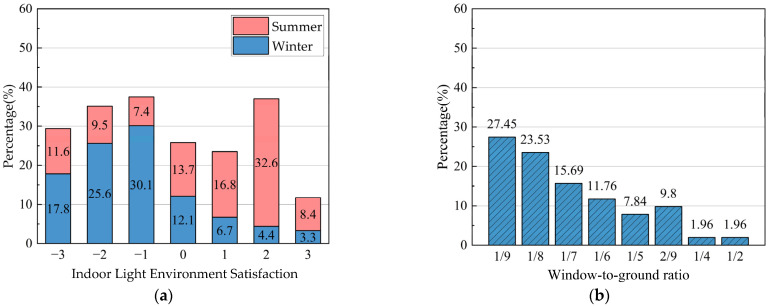
Results of the questionnaire survey on the indoor light environment of traditional houses in Xixinan Village. (**a**) Satisfaction distribution of indoor light environment. (**b**) Comparison of indoor windows and floor space in traditional residential houses.

**Figure 13 ijerph-20-03024-f013:**
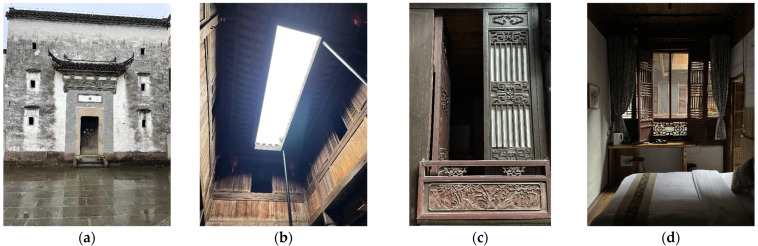
The interior light environment in “Yao’s House”. (**a**) Front and rear elevation; (**b**) patios; (**c**) windows; and (**d**) bedroom.

**Figure 14 ijerph-20-03024-f014:**
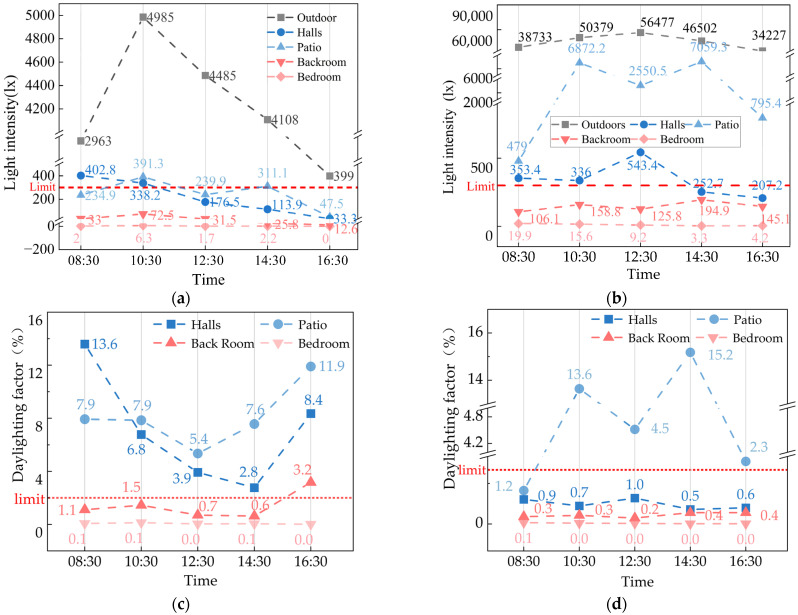
Indoor light environment monitoring results of “Yao’s House”. (**a**) Light intensity distribution in winter; (**b**) light intensity distribution in summer; (**c**) indoor daylighting in winter; and (**d**) indoor daylighting factor in summer.

**Figure 15 ijerph-20-03024-f015:**
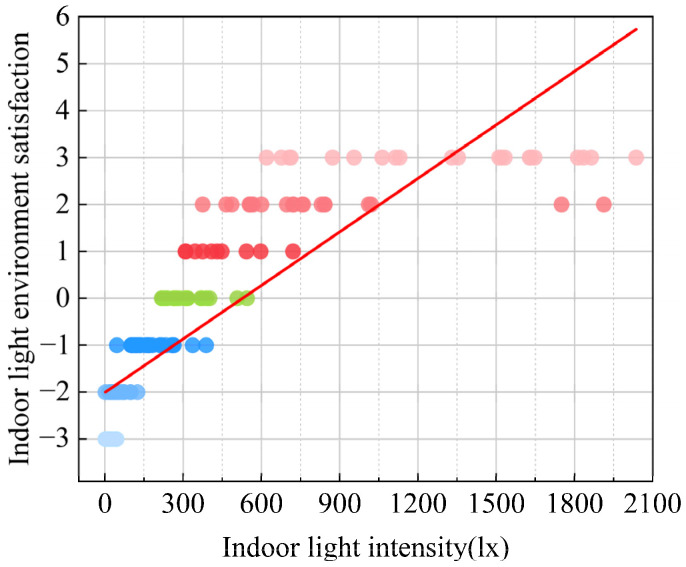
The relationship between indoor light intensity and satisfaction.

**Figure 16 ijerph-20-03024-f016:**
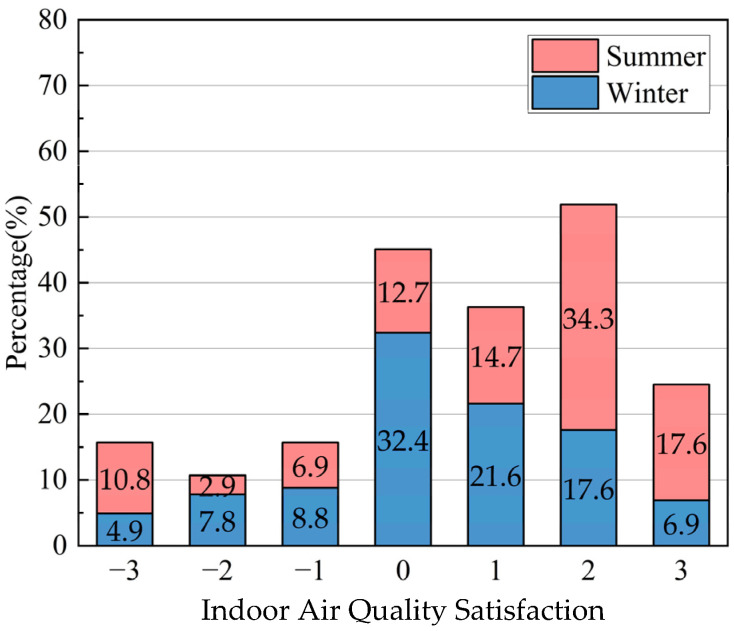
Indoor air quality satisfaction distribution of traditional houses in Xixinan Village.

**Figure 17 ijerph-20-03024-f017:**
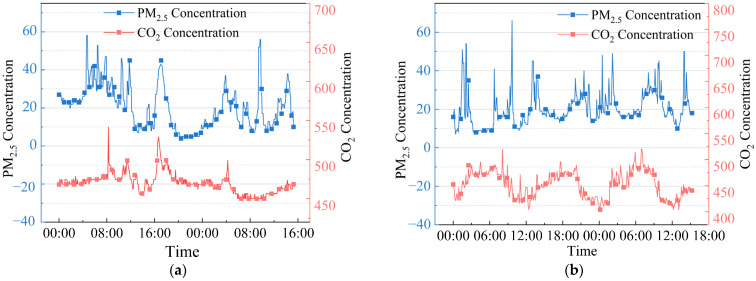
Distribution of CO_2_ concentration and PM_2.5_ concentration in indoor air quality. (**a**) Winter; and (**b**) summer.

**Figure 18 ijerph-20-03024-f018:**
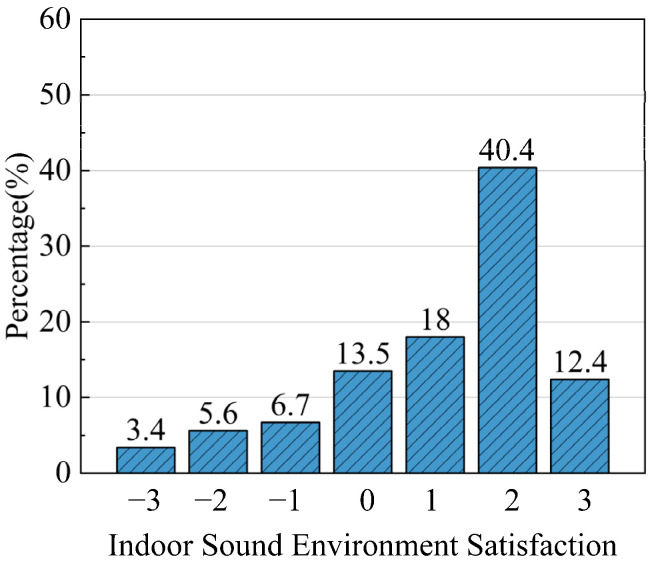
Indoor sound environment satisfaction distribution of traditional houses in Xixinan Village.

**Figure 19 ijerph-20-03024-f019:**
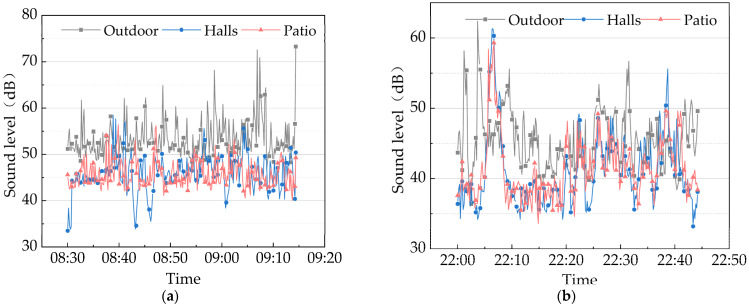
Sound pressure values of the “Yao’s House” at different test points. (**a**) Daytime; and (**b**) nighttime.

**Table 1 ijerph-20-03024-t001:** Hierarchy structure of indoor environmental comfort evaluation of traditional residential houses.

Target Layer	Criterion Layer	Index Layer	
A comprehensive evaluation of indoor environment quality and comfort of traditional residential buildings.	Thermal comfort	APMV [49]	Air temperature
			Air humidity
			Air velocity
			Global temperature
			Metabolic rate
			Clothing insulation
	Light comfort	Illuminance	
	Indoor air quality	PM_2.5_ concentration	
		CO_2_ concentration	
	Sound comfort	Sound level	

**Table 2 ijerph-20-03024-t002:** Basic information about residents.

Age	Sex		Total
	Male	Female	
<18	12 (48%)	13 (52%)	25 (6.9%)
18–30	17 (53.1%)	15 (46.9%)	32 (8.8%)
30–50	33 (40.7%)	48 (59.3%)	81 (22.5%)
50–70	69 (46.6%)	79 (53.4%)	148 (41.2%)
>70	33 (44.6%)	41 (55.4%)	41 (55.4%)
Total	164 (45.7%)	195 (54.3%)	359

**Table 3 ijerph-20-03024-t003:** Detailed information about the measuring device.

Name	Parameter	Range	Accuracy
Testo 175H1	Air temperature Relative humidity	−20~+55 °C 0~100%	±0.4 °C ±2%
Kestrel NK-5500	Wind velocity Wind direction	0.4~40 m/s	±4%
JTR-04	Globe temperature	−20~+80 °C	±0.5 °C
JTG-01	Illuminance intensity	0.1~100,000 lx	±4%
GT-1000	Particulate matter (PM)	0~99,999 μg/m^3^	±1%
Testo 535	CO_2_ concentration	0~10,000 ppm	±2%
TES1350A	A-weighted sound pressure level	35~130 dB	±2.0 dB

**Table 4 ijerph-20-03024-t004:** Reflection ratio of interior finishes.

Name	Halls		Back Room		Bedroom		Appropriate Value
Wall (Reflection ratio/material)	0.1	Dark-colored wood flooring	0.1	Dark-colored wood flooring	0.1	Dark-colored wood flooring	0.3~0.6
Ground (Reflection ratio/material)	0.23	Grey bricks	0.1	Dark-colored wood flooring	0.58	Light-colored wood flooring	0.1~0.5
Roof (Reflection ratio/material)	0.1	Dark-colored wood flooring	0.1	Dark-colored wood flooring	0.1	Dark-colored wood flooring	0.6~0.9

## Data Availability

Not applicable.

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
