# Peer review of "Indoor Environmental Comfort Assessment of Traditional Folk Houses: A Case Study in Southern Anhui, China"

_ijerph, 2023, doi:10.3390/ijerph20043024_

Round 1

Reviewer 1 Report (Previous Reviewer 2)

The authors improved a lot the paper, which is now acceptable even if I'm not still sure about its relevance in the scientific panorama.

My only final suggestion is to refine title, which is too long and descriptive.

Author Response

Dear Reviewer 1,

On behalf of all the co-authors, I would like to express my sincere appreciation for your valuable comments and professional suggestions on our work. All your comments are very helpful for improving our paper. Revised contents are marked in red in the revised manuscript. Below are our detailed reply and modifications:

Point 1: My only final suggestion is to refine title, which is too long and descriptive.

Response 1: Thank you very much for your professional comments on our work. In the revised version, we have changed the title of the paper to: Indoor Environmental Comfort Assessment of Traditional Folk Houses: A case study in southern Anhui, China

Please note that and check the modifications in the revised manuscript.

Once again, we sincerely appreciate your valuable comments and professional guidance on our work. We are looking forward to hearing from you again.

Yours sincerely,

Chao Pan (on behalf of the co-authors)

School of Architecture and Urban Planning, AHJZU, China.

Reviewer 2 Report (Previous Reviewer 3)

The manuscript has been significantly improved,I agree with its publication.

Author Response

Dear Reviewer 2,

On behalf of all the co-authors, I would like to express my sincere appreciation for your valuable comments and professional suggestions on our work. All your comments are very helpful for improving our paper.  

Yours sincerely,

Chao Pan (on behalf of the co-authors)

School of Architecture and Urban Planning, AHJZU, China.

This manuscript is a resubmission of an earlier submission. The following is a list of the peer review reports and author responses from that submission.

Round 1

Reviewer 1 Report

The article describes the situation of living in traditional residential houses in the region of Anlmi Province - China. Study was provided on the basis of direct measurement of temperature and light intensity. Besides that questionare survey was done to check objective filling of the population living in presented area. Then statistical analysis of obtained data was provided. 

8-10 Please make corrections of a numbering.

Please make Figures more readible.

Please put in the Section 4. Conclusions - subpoints to show the most important findings of the study, what will make article more readible.

Author Response

请参阅附件。

Reviewer 2 Report

The paper reports and investigation of environmental parameters inside historical buildings in a Chinese region. The topic can be interesting, but at present it does not looks like interesting for an international audience and its applicability is not clear outside the context. The authors should work on that. Moreover, I am not sure if the journal is suitable for the paper, or other MDPI journal would be better. For how the paper is present, something more focus on buildings would be correct rather than health. If editors and authors decide to keep it in the present journal, severe changes should be done in the background part, aiming at health perspective and not structural one. Moreover, three different buildings have been described, but it seems that measurements have been performed only in one type. Positive aspects are the very high number of questionnaires analyzed and high response rate. Conclusions should then be expanded in order to provide future perspectives to the work, as well as further interests and applicability to the present work. Other suggestions are reported to authors.

Abstract is too long.

Avoid use of we in formal English writing style.

Avoid repetitions of sentences in different part of the paper. Try changes something to make them appear different in the form at least.

2.3.4. is badly written and includes many errors. Please investigates better the argument.

The acoustic part, in all the work, is really treated marginally. Many more attention should be put on it. Please described the measurements taken, the length, position, duration, instrument and everything needed for the understanding. What parameters have been used? I would suggest to refer to room acoustic parameters to be considered, such as reverberation time, wall insulation, and so on.

A couple sentences in the introduction must also been added to balance the actual unbalance in the different arguments treated. A period investigating the importance of noise comfort and exposure is then needed, such as:

The health effect sentence in the introduction is definitively misleading as it is at present. Please provide major integration in that part by adding a plethora of more demonstrated effects. Among them: sleep disorders with awakenings (Capezuti, E., Pain, K., Alamag, E., Chen, X., Philibert, V., & Krieger, A. C. (2022). Systematic review: auditory stimulation and sleep. Journal of Clinical Sleep Medicine, 18(6), 1697-1709.), learning impairment (Minichilli, Fabrizio, et al. "Annoyance judgment and measurements of environmental noise: A focus on Italian secondary schools." International journal of environmental research and public health 15.2 (2018): 208), hypertension ischemic heart disease (Bolm-Audorff, U., Hegewald, J., Pretzsch, A., Freiberg, A., Nienhaus, A., & Seidler, A. (2022). Letter to the editor regarding," The effect of occupational exposure to noise on ischaemic heart disease, stroke and hypertension: A systematic review and meta-analysis from the WHO/ILO joint estimates of the work-related burden of disease and injury". Environment international, 161, 107104.), diastolic blood pressure (Petri, D., et al. Effects of Exposure to Road, Railway, Airport and Recreational Noise on Blood Pressure and Hypertension. Int. J. Environ. Res. Public Health 2021, 18(17), 9145), reduction of working performance Rossi, L., Prato, A., Lesina, L., & Schiavi, A. (2018). Effects of low-frequency noise on human cognitive performances in laboratory. Building Acoustics, 25(1), 17-33.), annoyance (Miedema HME, Oudshoorn CGM. Annoyance from transportation noise: relationships with exposure metrics DNL and DENL and their confidence intervals. Environ Health Perspect 2001; 109: 409–16;). Depending on the investigated area, there are different major noise sources responsible for the impact on on human life style: road traffic (Cueto, J. L., Petrovici, A. M., Hernández, R., & Fernández, F. (2017). Analysis of the Impact of Bus Signal Priority on Urban Noise. Acta Acustica united with Acustica, 103(4), 561-573., railway traffic (Bunn, Fernando, and Paulo Henrique Trombetta Zannin. "Assessment of railway noise in an urban setting." Applied Acoustics 104 (2016): 16-23;), airports (Iglesias-Merchan, Carlos, Luis Diaz-Balteiro, and Mario Soliño. "Transportation planning and quiet natural areas preservation: Aircraft overflights noise assessment in a National Park." Transportation Research Part D: Transport and Environment 41 (2015): 1-12; Gagliardi, P., Teti, L., & Licitra, G. (2018). A statistical evaluation on flight operational characteristics affecting aircraft noise during take-off. Applied acoustics, 134, 8-15.).”

Avoid bullets (1., 2.) in the conclusions. Most importantly, conclusions should better summarize all the work and provide further interests for other research. The present work does not look applicable and usable in other context, or interesting for scientific audience. Please try focus on put the work into a wider context of usability.

I believe the paper can be a research article if properly managed, and not a simple "Essay".

Reviewer 3 Report

Authors conducted an in-depth and comprehensive field study of traditional residential houses in Xixinan Village in southern Anhui, and determined the adjustment range of indoor environment parameters for residents' comfort needs. The research results provide an important theoretical basis for improving the indoor quality environment of traditional residential houses. The topic of the research is relevant to built environment and human comfort. However, to help improve the quality of this manuscript and make it outstanding among the published works, I have provided the following major comments:

1. Line 259: The authors listed the model and accuracy of the monitoring instrument. Is the accuracy the result of the author's field experiment? Please introduce the accuracy in detail. This would make it a more comprehensively research.

2. Lines 176-178: Why did the authors choose these factors (Index Layer in Figure 4)  evaluation thermal comfort?

3. Lines 474-482: The authors presented the CO2 and PM2.5 concentrations in indoor, but did not explain whether PM2.5 changed in different seasons. Similarly, the authors also mentioned that smoke exhaust facilities poor working efficiency resulting in the indoor pollutants cannot be quickly discharged to the outside. Have the author monitored changes in CO2 and PM2.5 during cooking? More detailed experiments can be better explain air quality varieties in indoor.

4. The part of “Results and Discussions”: Generally, the “Results” and “Discussions” should be the independent part, and I suggest that the authors could separate them. For example, Lines 301-306 and 341-362 are the discussion section.

5. The limitations of this research have not been mentioned in in the manuscript, I think these relevant contents should be mentioned by the authors.

Technical comments:

1. Line 431 This is wrong: “T This may be due to the fact...”. Please check the write errors carefully.

2. The figures numbers in the manuscript are not numbered in order, repeated or out of order. Please check and modify them.
